# Assessing Psychological Morbidity in Cancer-Unaffected *BRCA1/2* Pathogenic Variant Carriers: A Systematic Review

Anna Isselhard [1],* , Zoë Lautz [1], Kerstin Rhiem [2] and Stephanie Stock [1]

1 Institute of Health Economics and Clinical Epidemiology, University Hospital Cologne, 50935 Cologne, Germany
2 Center for Hereditary Breast and Ovarian Cancer and Center for Integrated Oncology (CIO), Medical Faculty, University Hospital, 50937 Cologne, Germany
* Correspondence: anna.isselhard@uk-koeln.de

**Abstract:** Female *BRCA1/2* pathogenic variant carriers have an increased lifetime risk for breast and ovarian cancer. Cancer-unaffected women who are newly diagnosed with this pathogenic variant may experience psychological distress because of imminent health threat. No comprehensible review on psychological morbidity in cancer-unaffected *BRCA1/2* pathogenic variant carriers is currently available. This review aims to give an overview about all available the studies in which psychological outcomes have been assessed in cancer-unaffected *BRCA1/2* pathogenic variant carriers, whether as a primary outcome or secondary measurement. A systematic search across four databases (Web of Science, PubMed, ScienceDirect, and EBSCO) was conducted. Studies had to report on cancer-unaffected pathogenic variant carriers (exclusively or separately) and use a validated measure of psychological morbidity to be eligible. Measures were only included if they were used in at least three studies. The final review consisted of 45 studies from 13 countries. Distress measures, including anxiety and cancer worry, were most often assessed. Most studies found a peak of distress immediately after genetic test result disclosure, with a subsequent decline over the following months. Only some studies found elevated distress in carriers compared to non-carriers in longer follow-ups. Depression was frequently investigated but largely not found to be of clinical significance. Quality of life seemed to be largely unaffected by a positive genetic test result, although there was some evidence that younger women, especially, were less satisfied with their role functioning in life. Body image has been infrequently assessed so far, but the evidence suggested that there may be a decrease in body image after genetic test result disclosure that may decrease further for women who opt for a prophylactic mastectomy. Across all the outcomes, various versions of instruments were used, often limiting the comparability among the studies. Hence, future research should consider using frequently used instruments, as outlined by this review. Finally, while many studies included cancer-unaffected carriers, they were often not reported on separately, which made it difficult to draw specific conclusions about this population.

**Keywords:** *BRCA1*; *BRCA2*; breast cancer; anxiety; distress; cancer worry; patient experience

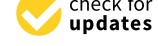



## 1. Introduction

*BRCA1* and *BRCA2* are tumor suppressor genes that encode proteins, which are responsible for repairing disruptions in damaged DNA that could otherwise result in tumor formation [1,2]. Inheriting a pathogenic variant in either of the two genes leads to erroneous DNA repair and, subsequently, a high risk for breast and ovarian cancer in women [1–3]. For breast cancer, the lifetime risk is roughly five to seven times higher for *BRCA1/2* pathogenic variant carriers compared to women in the general population [3]. For ovarian cancer, the risk is roughly 20 times higher for *BRCA2* and 40 times higher for *BRCA1* pathogenic variant carriers [3]. Albeit independent, a pathogenic variant in either gene is inherited from parent to offspring in autosomal dominant heredity. Therefore, cancer-unaffected members of families with a known *BRCA1/2* pathogenic variant are generally

offered genetic counseling and testing [4]. Likewise, index patients of families with a high incidence of breast and ovarian cancers with unknown pathogenic variant status may be offered genetic counseling and testing based on a familial risk assessment. Upon reasonable probability of carrying a pathogenic variant, a blood sample is preferentially drawn from a cancer patient (index patient) and tested. The test result may be positive (individual is a *BRCA1/2* pathogenic variant carrier), negative (individual is a *BRCA1/2* pathogenic variant non-carrier in a *BRCA1/2*-positive family), non-informative (no pathogenic variant was detected in a particular gene), or inconclusive (no pathogenic variant in *BRCA1/2* was found, but a variant of unknown significance (VUS) was) [5]. Pathogenic variant carriers are confronted with difficult decisions in the case of a positive genetic test result on how to deal with their personal cancer risk. Women without previous breast or ovarian cancer history have to make difficult decisions on which risk-reducing strategy to adopt. For breast cancer, this may mean a risk-reducing bilateral mastectomy or participation in intensified surveillance programs [6–8]. While a risk-reducing bilateral mastectomy may reduce breast cancer incidence for carriers of both pathogenic variants, as well as mortality for *BRCA1* pathogenic variant carriers [8,9], worsening of body image and sexual satisfaction have been reported, even with immediate reconstruction [10–13]. On the other hand, breast surveillance is less invasive and can provide survival benefits [14] but cannot reduce breast cancer risk. Both of these options might, therefore, induce distress and worsen psychological wellbeing, as both options come with significant downsides [15]. For ovarian cancer, the only option for effective risk management is a prophylactic bilateral salpingo-oophorectomy [16–18]. For surgical options in particular, female carriers must decide whether to opt for them at all and at what point in their life depending on age-dependent risk, since surgical procedures impact the possibility of bearing or breastfeeding children.

Consequently, undergoing genetic testing, receiving a positive genetic test result, and sharing the test results friends and families may influence levels of psychological morbidity [19,20]. Some women go as far as describing genetic test result disclosure as traumatic [21]. Various studies have assessed psychological wellbeing and morbidity in *BRCA1/2* pathogenic variant carriers, both qualitatively [19,21–24] and quantitatively [25–27]. Previous reviews have attempted to condense the evidence available [12,20,26,28–31]. However, these reviews (1) have focused on the efficacy of psychosocial interventions [28,29], (2) have focused on the psychological effects of different risk-management strategies [12,31], (3) have only included cancer-affected *BRCA1/2* carriers [30], or (4) have reported men and women or cancer-unaffected and cancer-affected *BRCA1/2* pathogenic variant carriers combined [20,26]. This is problematic, as there appears to be a non-negligible difference between cancer-affected compared to cancer-unaffected pathogenic variant carriers [26,32].

To the best of our knowledge, no comprehensive systematic review about the psychological morbidity that female cancer-unaffected *BRCA1/2* pathogenic variant carriers experience after genetic test result disclosure is available thus far. Therefore, the aim of this review is to fill this gap in the literature and explore the short- and long-term psychological consequences of receiving a positive genetic test result for *BRCA1* or *BRCA2* in women without a personal cancer history. To reach these aims, this review sets out to answer two questions:

- How is the psychological morbidity in cancer-unaffected *BRCA1/2* pathogenic variant carriers, both immediately after genetic test result disclosure and long-term?
- Which instruments are frequently employed to assess these psychological morbidities?

## 2. Materials and Methods

The 2020 Preferred Reporting Items for Systematic Reviews and Meta-Analyses (PRISMA) guidelines were utilized for this review [33]. Four bibliographic databases (Web of Science, PubMed, ScienceDirect, and EBSCO) were systematically searched for studies published from 1997 to January 2023. The search terms included the following keywords, and PubMed medical subject headings (MeSHs) were included individually and in combination depending on the database: *BRCA*, *BRCA1/2*, psychosocial impact,

psychosocial distress, coping, anxiety, depression, mental health, psychological adjustment, and mental disorder. The review was not prospectively registered, but the authors will provide protocol upon request.

### 2.1. Eligibility Criteria

Studies were deemed eligible if they were written in English and if they fulfilled the criteria, as determined by the PICOS framework [34,35].

- Participants: the review focused on cancer-unaffected female adults (age ≥ 18 years) with a confirmed pathogenic variant in either *BRCA1* or *BRCA2*.
- Intervention: no special intervention was specified.
- Comparison: studies that compared BRCA pathogenic variant carriers with women who received negative or inconclusive BRCA genetic test results, as well as studies that compared cancer-affected vs. cancer-unaffected pathogenic variant carriers were also included.
- Outcomes: the review included short-term and long-term psychological consequences that were measured with validated instruments.
- Study design: only quantitative studies, irrespective of study design (randomized or non-randomized trials, longitudinal cohort, cross-sectional, or case control), were included; qualitative studies were excluded from the present review.

### 2.2. Exclusion Criteria

Exclusion criteria consisted of studies not written in English, books, qualitative studies, literature reviews, case reports, or letters to the editor. Studies were also excluded if there was no reporting of psychological consequences or if studies did not specifically identify the population as (1) female, (2) cancer-unaffected, and (3) definitive *BRCA1/2* pathogenic variant carriers. Therefore, studies grouping results for cancer-unaffected with cancer-affected pathogenic variant carriers, carriers with non-carriers, or female with male carriers or those not defining the pathogenic variant as BRCA1/2 were excluded. Additionally, to provide the most value, studies were only included if they measured psychological morbidity with a validated questionnaire that at least three studies used.

### 2.3. Data Extraction, Data Synthesis, and Quality Assessment

After removal of duplicates, a stepwise approach was undertaken: first, two authors screened titles and abstracts independently (AI and ZL). Conflicts in screening were resolved by discussion. If the disagreement could not be solved quickly, the record went through a full-text review. Next, two authors (AI and ZL) independently screened the full-text articles. Disagreements during this process were solved by discussion. The included studies were analyzed according to the predefined PICOS criteria (see Section 2.1). For each included study, one author (ZL) extracted the following information: full reference, study design, duration of follow-up, and participant characteristics (sample size, age, *BRCA1/2* pathogenic variant status, and psychological outcome). Information extraction was overseen and quality-controlled by one author (AI). The findings were divided and grouped into the outcomes utilized within the studies. The goal of this review was a descriptive data analysis and synthesis of evidence. We, therefore, clustered outcomes with their respective validated instruments.

The quality of the included studies was assessed with the AXIS tool [36]. This tool was developed for non-experimental research and includes 20 discrete-choice questions that may be answered with yes or no (e.g., "Was the target population clearly defined?" or "Was ethical approval or consent of participants attained?"). Two reviewers (AI and ZL) rated each item independently and resolved disagreements in the process via discussion. A point was assigned for an item if methodological quality was met, resulting in a score from 0 to 20 for each study, with higher scores indicating higher study quality. The full AXIS assessment can be found in Supplementary Material File S1.

### 3. Results

The full flow-chart for the review process is displayed in Figure 1. The initial search yielded 810 records. After duplicates were removed, 478 records were screened for eligibility, and 264 records went through full-text review. Additionally, five records were identified by hand search. Forty-five studies met the eligibility criteria and were included in this review. The total number of participants from all the included studies was *n* = 2442, with an age range of 18–83. Overall, the studies showed good quality (see quality assessment 3.5), with some exceptions. Some studies only partially reported outcomes separately for cancer-affected versus cancer-unaffected pathogenic variant carriers. All the studies included in the review are shown in Table 1.

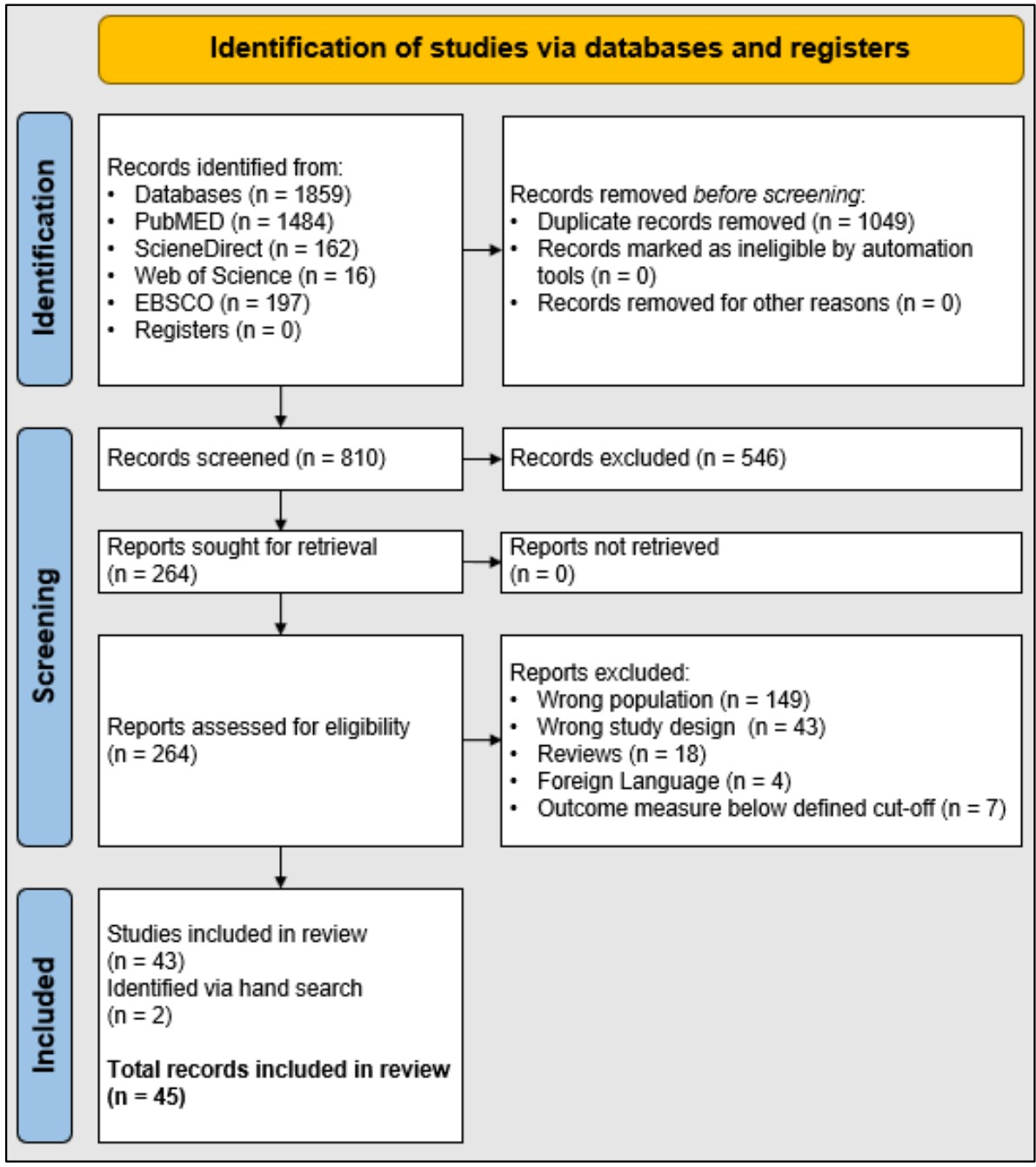

**Figure 1.** PRISMA flow chart for the identification of studies.

**Table 1.** Descriptive data of the studies (*n* = 45) and measures used.

| First Author, Year | Country | Participant Characteristics [1] | Study Design, Follow-Up Length | BHS | BIQ | BSI | CES-D | CWS | HADS | IES | GHQ | SF-12/36 | STAI |
|---|---|---|---|---|---|---|---|---|---|---|---|---|---|
| Borreani et al., 2014 [37] | Italy | *n* = 27 Age range: 26–75 | Longitudinal, 15 months | | | | | √ | √ | | | √ | |
| Brand et al., 2021 [38] | Germany | *n* = 48 Age mean: 40 | Cross-sectional | | | | | √ | | | | | |
| Buchanan et al., 2017 [39] | U.S. | *n* = 97 Age range: 25–40+ | Cross-sectional | | | | | | | √ | | √ | √ |
| Carpenter et al., 2014 [40] | U.S. | *n* = 26 Age mean: 42.9 | Experimental | | | | | | | √ | | | |
| Claes et al., 2005 [41] | Belgium | *n* = 34 Age range: 19–61 | Longitudinal, 1 year | | | | | | | √ | | | √ |
| Croyle et al., 1997 [42] | U.S. | *n* = 13 Age range: 19–83 | Longitudinal, 2 years | | | | | | | √ | | | √ |
| Dagan and Gil, 2004 [43] | Israel | *n* = 36 Age mean: 54.1 | Retrospective | | | | √ | | | | | | |
| Dagan and Shochat, 2009 [44] | Israel | *n* = 17 Age mean: 51.4 | Case control | | | | √ | | √ | | | √ | |
| Dorval et al., 2006 [45] | Canada | *n* = 19 Age mean: 48 | Longitudinal, 36 months | | | | | | | √ | | | |
| Ertmanski et al., 2009 [46] | Poland | *n* = 56 Age range: 18–56+ | Longitudinal, 1 year | | | | | | | √ | | | √ |
| Finch et al., 2013 [47] | Canada | *n* = 59 Age range: 35–69 | Longitudinal, 1 year | | | | √ | | | √ | | √ | |
| Foster et al., 2007 [48] | U.K. | *n* = 53 Age range: 23–72 | Longitudinal, 3 years | | | | | | √ | | √ | | |
| Geirdal and Dahl, 2008 [49] | Norway | *n* = 68 Age mean: 42 | Cross-sectional | | | | | | √ | | | | |
| Geirdal et al., 2005 [50] | Norway | *n* = 68 Age mean: 42 | Cross-sectional | √ | | | | | | √ | √ | √ | |
| Gopie et al., 2013 [51] | Netherlands | *n* = 44 Age mean: 37.1 | Longitudinal, 21.7 months | | √ | | | | | √ | | √ | |
| Graves et al., 2012 [52] | U.S. | *n* = 47 Age mean: 54.1 | Longitudinal, 5 years | | | | √ | | | √ | | | √ |

**Table 1.** *Cont.*

| First Author, Year | Country | Participant Characteristics [1] | Study Design, Follow-Up Length | BHS | BIQ | BSI | CES-D | CWS | HADS | IES | GHQ | SF-12/36 | STAI |
|---|---|---|---|---|---|---|---|---|---|---|---|---|---|
| Isern et al., 2008 [53] | Sweden | *n* = 27 Age range: 25–51 | Longitudinal, 42 months | | | | | | √ | | | √ | |
| Isselhard et al., 2023 [54] | Germany | *n* = 130 Age range: 24–60 | Cross-sectional | | | | | | √ | √ | | | |
| Julian-Reynier et al., 2010 [55] | France | *n* = 244 Age range: <30–50+ | Longitudinal, 60 months | | √ | | √ | | | √ | | | |
| Kinney et al., 2005 [56] | U.S. | *n* = 19 Age range: <40–50+ | Longitudinal, 1 year | | | | √ | √ | | √ | | | √ |
| Landau et al., 2015 [57] | Israel | *n* = 56 Age mean: 49.6 | Intervention, 12 weeks | | | √ | | √ | | | | | |
| Lapointe et al., 2013 [58] | France | *n* = 221 Age range: 20–60 | Longitudinal, 2 years | | | | √ | | | √ | | | |
| Lodder et al., 2001 [59] | Netherlands | *n* = 25 Age range: 19–68 | Longitudinal, 1–3 weeks | | | | | | √ | √ | | | |
| Lodder et al., 2002 [60] | Netherlands | *n* = 26 Age mean: 38.8 | Longitudinal, 12 months | | √ | | | | √ | √ | | | |
| Low et al., 2008 [61] | U.S. | *n* = 7 Age mean: 44.7 | Longitudinal, 6 months | | | | | | | √ | | | |
| Madalinska et al., 2007 [62] | Netherlands | *n* = 160 Age range: 35–50+ | Longitudinal, 12 months | | | | | √ | | √ | | √ | |
| Maheu et al., 2012 [63] | France | *n* = 217 Age range: <35–50+ | Longitudinal, 2 years | | | | √ | | | √ | | | |
| Maheu et al., 2014 [64] | France | *n* = 232 Age mean: 40.7 | Longitudinal, 12 months | | | | √ | | | √ | | | |
| Meiser et al., 2002 [65] | Australia | *n* = 30 Age mean: 40 | Longitudinal, 12 months | | | | | | | √ | | | √ |
| Metcalfe et al., 2012 [66] | Canada | *n* = 22 Age range: 25–70 | Longitudinal, 2 years | | | | | | | √ | | | |
| Metcalfe et al., 2017 [67] | Canada | *n* = 150 Age range: 25–60 | RCT, 12 months | | | | | | | √ | | | |
| Metcalfe et al., 2020 [68] | Canada | *n* = 576 Age range: 25–55 | Cross-sectional | | | | | | | √ | | | |

**Table 1.** *Cont.*

| First Author, Year | Country | Participant Characteristics [1] | Study Design, Follow-Up Length | BHS | BIQ | BSI | CES-D | CWS | HADS | IES | GHQ | SF-12/36 | STAI |
|---|---|---|---|---|---|---|---|---|---|---|---|---|---|
| O'Neill et al., 2009 [69] | U.S. | *n* = 14 Age range: 27–68 | Longitudinal, 1 year | | | | | | | √ | | | |
| Reichelt et al., 2004 [70] | Norway | *n* = 80 Age mean: 43.9 | Longitudinal, 6 weeks | √ | | | | | √ | √ | √ | | |
| Reichelt et al., 2008 [71] | Norway | *n* = 58 Age mean: 45.4 | Longitudinal, 18 months | √ | | | | | √ | √ | | | |
| Schwartz et al., 2002 [72] | U.S. | *n* = 35 Age mean: 45 | Longitudinal | | | | | | | √ | | | |
| Shochat and Dagan, 2010 [73] | Israel | *n* = 17 Age mean: 51.4 | Cross-sectional | | | | √ | √ | | | | | |
| Smith et al., 2008 [74] | U.S. | *n* = 20 Age range: 22–70 | Longitudinal, 6 months | | | | √ | | | √ | | √ | √ |
| Spiegel et al., 2011 [75] | U.S. | *n* = 51 Age range: 25–60 | Longitudinal, 6 months | | | | | √ | √ | | | | |
| Van Dijk et al., 2006 [76] | Netherlands | *n* = 22 Age range: <30–50+ | Longitudinal, 6 months | | | | | | √ | √ | | | |
| Van Egdom et al., 2020 [77] | Netherlands | *n* = 96 Age mean: 41.4 | Cross-sectional | | | | | | √ | | | | |
| Van Oostrom et al., 2003 [78] | Netherlands | *n* = 23 Age mean: 41.9 | Longitudinal, 4–6 years | | | √ | | | √ | √ | √ | | |
| Van Oostrom et al., 2007 [79] | Netherlands | *n* = 49 Age mean: 42.3 | Longitudinal, 12 months | | | | | | √ | √ | | | |
| Van Roosmalen et al., 2004 [80] | Netherlands | *n* = 68 Age mean: 37.6 | Longitudinal, 2 weeks | | | | | √ | | √ | | | √ |
| Watson et al., 2004 [81] | U.K. | *n* = 91 Age range: 23–72 | Longitudinal, 12 months | | | | | | √ | √ | √ | | |

[1] Sample size *n* refers to number of cancer-unaffected female *BRCA1/2* carriers in the sample and does not represent total sample size.

### 3.1. Study Characteristics

The studies were published from 1997 to January 2023. Most studies included participants from the U.S. (10 studies), the Netherlands (9 studies), Canada (5 studies), Norway, France, and Israel (4 studies each). Other countries included Italy, Belgium, Poland, the U.K., Sweden, France, and Australia. Most studies employed a (prospective) longitudinal cohort design (thirty-two studies), ranging in follow-up from one week to six years after test result disclosure. Additionally, studies with cross-sectional designs (eight studies) and randomized controlled intervention designs (two studies), as well as one experimental, one retrospective, and one case-control study, were included. Study populations had high heterogeneity in their sample sizes, from $n = 7$ to $n = 576$ cancer-unaffected pathogenic variant carriers. The age ranges in the studies were between 18 and 83 years old. In total, 11 measures were examined within this review (see Table 2).

**Table 2.** General outcomes and respective measures included in this review.

| General Outcome | Specific Measure |
|---|---|
| Distress | Impact of Event Scale (IES) [82,83] |
| | Hospital Anxiety and Depression Scale (HADS) [84] |
| | Cancer Worry Scale (CWS) [85,86] |
| | Spielberger State-Trait Anxiety Inventory (STAI) [87] |
| | Brief Symptom Inventory (BSI) [88] |
| | General Health Questionnaire (GHQ) [89] |
| Depression | Hospital Anxiety and Depression Scale (HADS) [84] |
| | Beck's Hopelessness Scale (BHS) [90] |
| | Center for Epidemiologic Studies Depression Scale (CES-D) [91] |
| Other | Short Form Health Survey (SF-36/SF-12) [92,93] |
| | Body Image Questionnaire (BIQ) [60] |

### 3.2. Distress Measures

The anxiety subscale of the Hospital Anxiety and Depression Scale (HADS) [84], the Cancer Worry Scale (CWS) [85,86], the Spielberger State-Trait Anxiety Inventory (STAI) [87], the Brief Symptom Inventory (BSI) [88], the General Health Questionnaire (GHS-28) [89], and the Impact of Event Scale (IES) [82] were characterized as psychological distress parameters.

#### 3.2.1. Impact of Event Scale

By far the most used questionnaire to measure distress was the IES, which was used by 34 studies [39–42,45–47,50–52,54–56,58–72,74,76,78–81]. The IES consists of two subscales for intrusion and avoidance. The revised IES-R additionally has a hyperarousal subscale. Twenty-eight studies used the original version of the questionnaire (IES), whereas three studies used the revised version (IES-R) [54,61,79], and three studies used the intrusion subscale only [62,70,71]. While cut-off values have been reported for different populations, they vary by version used and have been criticized for providing little clinical significance. Fourteen studies report higher IES scores in carriers compared to non-carriers within six months of test result disclosure [42,45,58,59,61,63,65,69,70,72,74,76,80,81]. Of these studies, five reported that distress remained significantly higher in carriers for up to one year after disclosure [58,65,72,76,81]. Contrarily, one study reported that, while carriers experienced higher distress immediately after genetic test result disclosure, there was no significant difference in the distress of non-carriers after 6 months [63]. Two long-term follow-ups with an average time of five years since genetic test result disclosure similarly found no difference between carriers and non-carriers [52,78]. One study reported that, even though distress was higher in carriers, carriers experienced a decrease from before to immediately after test result disclosure, indicating that knowing the test result regardless of the outcome may provide relief [42]. However, this was the only study with this particular result. In fact, four other studies found increases from before to immediately after test result disclosure in carriers [60,65,66,79]. Longitudinal studies among carriers suggested a de-

crease in distress anywhere between 6 months and two years after disclosure [60,66,67,79]. Higher distress was associated with higher adherence to recommendations about risk-reducing strategies [39] and, among those strategies, higher likelihood to opt for a bilateral mastectomy [60] or a salpingo-oophorectomy [55,62]. Five other studies reported significant decreases in distress after undergoing such risk-reducing surgeries [47,51,66,68,79]. Higher scores were significantly associated with general psychological distress [54] and with receiving a psychological consultation [64], providing some evidence for the real-world validity of the IES. In terms of validity, however, one author pointed to the importance of the definition of the "event" in question: test result disclosure or cancer itself [45]. In fact, one study found differences between carriers and non-carriers in distress when the IES was framed for ovarian cancer but not when it was framed for breast cancer [41]. Therefore, precise wording is important for the interpretation and comparability of results.

### 3.2.2. Hospital Anxiety and Depression Scale—Anxiety Subscale

Twelve studies assessed anxiety with the anxiety subscale of the HADS (HADS-A) [37,49,50,53,54,59,60,70,71,75,77,78]. Among the studies that compared anxiety in carriers with anxiety in non-carriers, two studies reported differences [59,60]. Both studies were written by the same authors and presumably reported on the same population, with one being focused on anxiety 1–3 weeks after test result disclosure [59] and the other being a one year follow-up [60]. The first of the two studies showed that non-carriers experienced a reduction in anxiety from before genetic testing to shortly after test result disclosure, whereas carriers showed an increase in anxiety [59]. A subgroup analyses based on high and low baseline anxiety was performed and identified a diverging pattern of results: pathogenic variant carriers with high pre-test anxiety remained highly anxious after receiving test results, whereas non-carriers with high pre-test anxiety showed a decrease in anxiety. Further, pathogenic variant carriers with low pre-test anxiety showed an increase in anxiety, whereas non-carriers with low pre-test anxiety showed unchanged levels of anxiety post-test. The second study showed that, at 1 year after receiving test results, anxiety levels for carriers and non-carriers were similar and that those with clinically high scores shortly after test result disclosure remained anxious at 1 year after disclosure [60]. Indeed, many studies found that pre-test anxiety levels were a good predictor of anxiety longitudinally [37,60,75,78]. One study specifically showed that, even after 5 years, current anxiety was best predicted by anxiety pregenetic test result disclosure, regardless of carrier status [78]. The authors of this particular study reported that anxiety in carriers spiked to just sub-clinical levels right after genetic test result disclosure but returned to the level of non-carriers after six months. They noted an increase in anxiety 5 years after genetic post-result disclosure that was present for carriers and non-carriers alike [78]. In contrast to these findings, two other studies found scores well below the clinical threshold for carriers and showed that anxiety scores in carriers were lower compared to women from high-risk families with an absence of demonstrated pathogenic variants [49,50]. Roughly half of the studies included percentages of potential clinical cases (HADS-A score $\geq$ 8) and reported that roughly one-in-four to one-in-five-carriers (19–24%) showed clinical anxiety [49,53,54,59,60,70,71]. One study reported almost half (49%) of participants scoring in the clinical anxiety range [75]. However, this higher occurrence may have been found because the sample in this study consisted of carriers who were or were not recalled after a suspicious MRI report in intensified breast cancer screening. The anxiety might, therefore, be a result of this recall and not of the genetic test result itself, as the recalled group showed significantly higher anxiety than the non-recalled group. It was shown that, even among recalled carriers, the scores returned to below baseline 6 months after genetic test result disclosure. Three studies were identified that compared carriers opting for different preventive options (risk-reducing surgery vs. surveillance) [37,60,77]. One study reported scores on the higher end of the normal range for both women who opted for surveillance and women who opted for surgery, with women in the surveillance group showing marginally, but not significantly, higher scores [37]. Another study showed a contrary result, with carriers

who opted for prophylactic mastectomy showing significantly higher anxiety compared to carriers opting for surveillance [60]. A reduction in anxiety from immediately after test result disclosure to 1 year after result disclosure was reported regardless of preventive option but was steeper for women who opted for a mastectomy. Finally, one study compared women opting for surveillance or bilateral prophylactic mastectomy with immediate breast reconstruction and found no difference in anxiety between the two [77].

### 3.2.3. Cancer Worry Scale

Thirteen studies measured cancer worry utilizing different versions of the CWS [37,38,44,48,56,57,62,73,75,76,78,79,81]. There was high heterogeneity in the versions used, with one study using a single-item version [76], two studies using a three-item version [37,56], five studies using a four-item version [38,44,73,75,79], two studies using a five-item version [62,78], and two studies using a revised six-item version [48,81]. One study did not specify which version was used [57]. Two studies did not report results relevant to the population [56,57]. Three studies identified no difference in cancer worry in carriers when compared to non-carriers from high-risk families [48,73,78]. In contrast, eight studies identified increased cancer worry, each with unique comparators [37,38,44,62,75,76,79,81]. Three studies compared carriers with non-carriers from high-risk families and found higher cancer worry in those with pathogenic variants for up to one year after genetic test result disclosure [44,79,81]. One of these studies further specified that, especially, carriers under the age of 35 experienced higher levels of cancer worry compared to carriers over 50 years one month after genetic test result disclosure [81]. This difference, however, was no longer significant at one year after genetic test result disclosure. This may be indicative of the complexity of decision making in premenopausal women immediately after genetic test result disclosure. Two other studies provided additional evidence for this by displaying an increase in cancer worry for up to one month after disclosure, with a subsequent decline in cancer worry at six months after genetic test result disclosure [76,79]. Another study compared carriers opting for different preventive strategies (prophylactic surgery vs. intensified breast cancer screening) [62]. The results revealed that, specifically, the surgery group showed an increase in cancer worry symptoms. Further, in another study [75] there was an increase in cancer worry over time. However, the study compared carriers who were recalled after a first MRI with women who were not recalled. Although there was no difference in cancer worry symptoms at the first MRI appointment, there was a significant increase in cancer worry symptoms in the recalled group. The non-recalled group did not exhibit this pattern, indicating that imminent cancer diagnosis may be relevant to the genesis of higher cancer worry.

### 3.2.4. Spielberger State-Trait Anxiety Inventory

Nine studies measured anxiety using the STAI [39,41,42,46,52,56,65,74,80], eight of which used the state anxiety subscale only. Only one study used both the state and the trait subscales [46]. While this study found no increase in state anxiety right after result disclosure, as well as at one year after, women with the highest trait anxiety also experienced the highest spike in state anxiety after genetic test result disclosure [46]. No specific outcome was reported in one study [52]. Three studies found that non-carriers experienced significantly less state anxiety after genetic test result disclosure, whereas carriers remained at a stable level or experience slightly more anxiety [41,56,65]. In fact, two studies found significantly higher state anxiety in carriers compared to non-carriers at 1–2 weeks after genetic test result disclosure [42,80]. Another study found higher state anxiety at three months after genetic test result disclosure in carriers compared to non-carriers but no longer at six months [74]. Likewise, another study found no differences between carriers and non-carriers at 4 months and 12 months after genetic test result disclosure [65]. One study found that anxiety was not related to adherence to recommended risk management [39].

### 3.2.5. Brief Symptom Inventory

Six studies assessed psychological distress via the BSI [43,44,47,52,57,73]. Different versions were used, with 1 study utilizing the 53-item version [43], 2 studies utilizing the 48-item version [44,73], 2 studies utilizing the 18-item version [47,57], and 1 study using the anxiety subscale only [52]. For one study, no outcome was specified [52]. Scoring schemes and subsequent cut-off values varied depending on the version used. One study found subclinical levels in carriers after genetic test result disclosure and no change at a three month follow-up [57]. Two studies did not identify an increase in distress in carriers compared to non-carriers from high-risk families [44,57,73]. Conversely, two other studies found that the scores of the somatization subscale were increased in carriers compared to non-carriers of all age groups [43], as well as in premenopausal carriers compared to postmenopausal carriers [47]. High scores on the somatization subscale represent a high focus on physical dysfunction (e.g., pain, fatigue, dizziness, numbness, or tingling) that may, in turn, cause psychological distress. Identifying differences in psychological distress was not related to the version of the BSI used.

### 3.2.6. General Health Questionnaire

Four studies assessed generalized psychological distress via the GHQ-28 [48,50,70,81]. A score $\geq 5$ indicates clinically significant distress [89]. All the included studies reported below this cut-off score, albeit some only marginally [48,81]. Two studies found lower psychological distress in identified carriers compared to untested members of high-risk families [50,70]. Two other studies reported an increase in psychological distress from before genetic testing to 12 months [81] or 3 years after result disclosure [48]. Even though the reported means did not tangent the cut-off score, one of these studies reported that almost 20% of the study participants scored above the cut-off score three years after genetic test result disclosure [48]. Another study identified that carriers aged 35–49 experienced significantly higher psychological distress than high-risk non-carriers 1 month after genetic test result disclosure [81].

### 3.2.7. Summary Distress Outcomes

In conclusion, many studies found a slight elevation in distress outcomes shortly after genetic test result disclosure. The majority of the studies reported that up to one-fourth of carriers experienced symptoms of anxiety disorder after genetic test result disclosure, irrespective of the instrument used. Longitudinal studies suggested that, even though anxiety symptoms peaked after genetic test result disclosure, they usually declined to the level of non-carriers over time. However, carriers with high pre-test anxiety may experience clinical anxiety, even at longer follow-ups. Some studies provided limited evidence for age dependence, with younger women showing higher distress than older women, especially immediately after genetic test result disclosure. Furthermore, there was some degree of evidence to suggest that those with higher distress were more likely to opt for surgery, albeit the causative nature of this relationship remains unclear. Therefore, sensitive screening tools to identity this subgroup may be beneficial to alleviate long-term distress and prevent the manifestation of anxiety disorders. Finally, some studies showed that carriers showed lower anxiety compared to untested women, suggesting that receiving a definitive test result, regardless of if a pathogenic variant was in fact found, may provide a relief in anxiety.

### 3.3. Depression

The depression subscale of the Hospital Anxiety and Depression Scale (HADS-D) [84], the Center for Epidemiologic Studies Depression Scale (CES-D) [91], and the Beck Hopelessness Scale (BHS) [90] were characterized as measures of depression.

### 3.3.1. Hospital Anxiety and Depression Scale—Depression Subscale

Ten studies assessed depression with the HADS-D [37,50,53,54,59,70,71,75,77,78]. Analogous to the anxiety subscale, a score ≥8 indicates signs of clinical depression. All the studies reported means well below this cut-off. Among the studies that reported a percentage of cases, the numbers ranged from 2–12.5% of possible clinical depression cases, indicating that depression was as prevalent as in the general population [50,53,54,59,75]. In fact, two studies found that the depression scores in carriers were significantly lower than in the healthy population [50,70]. One of these studies compared collected data from carriers with published normative data [70], whereas one study simultaneously collected data from carriers, non-carriers, and controls and found that carriers had fewer depressive symptoms compared to the other two groups [50]. Furthermore, two other studies found no difference between carriers and non-carriers in terms of depression [53,78]. One study that compared carriers and non-carriers from before to after genetic test result disclosure found an increase from before to after genetic test result disclosure for carriers and the opposite effect for non-carriers [59]. Two studies compared the depression scores of women opting for surveillance vs. risk-reducing surgeries and found no difference [37,77]. Finally, one study found that carrier depression scores were not affected by recall after a suspicious MRI [75], suggesting that depression was not influenced by imminent danger of cancer.

### 3.3.2. Center for Epidemiologic Studies Depression Scale

Seven studies assessed depression utilizing the CES-D [55,56,58,63,64,74,80], of which all but two reported outcomes relevant to the population [56,64]. Different cut-off scores have been put forward, ranging from scores ≥16 to ≥23 indicating a clinical case of depression. One study identified higher depressive symptoms in carriers compared to the general female population at baseline, with 21.3% of women scoring in the clinical depression range (scores ≥ 23) [55]. Three studies found increases from before to after genetic test result disclosure [58,74,80]. One of these studies reported means over the cut-off score of 16 at one week and three months after genetic test result disclosure, with no difference between carriers and non-carriers [74]. Similarly, another study found no differences between carriers and non-carriers but identified an increase in depression from pre-test result disclosure to 15 days after, with a subsequent decrease to pre-test levels after one year [58]. One study looked at risk management behaviors and found that women with fewer depressive symptoms were more likely to conduct regular breast self-examination [63].

### 3.3.3. Beck Hopelessness Scale

Three studies assessed hopelessness and associated suicidal ideation using the BHS [50,70,71]. A score between 4 and 8 generally indicates mild hopelessness, whereas a score of ≥9 suggests more severe hopelessness that predicts the presence of at least some suicidal ideation. Two of the studies reported mean scores in the higher end of the normal range [50,70]. One study did not specify the mean for the sample but reported a significant association to psychological distress in general [71].

### 3.3.4. Summary Depression Outcomes

The patterns of the results from these depression measures suggested that carriers did not show increased depressive symptoms following test disclosure, and some studies remarkably even identified levels of depression that were lower than those in the normal population. Of these depression measures, the CES-D appeared to be the most sensitive in detecting depression in *BRCA1/2* carriers. However, even studies using this instrument showed that depressive symptomology decreased over time, and no lasting effects were found. Only one study showed that depression scores remained above pre-test levels for up to two years. Studies using the other questionnaires indicated that hopelessness or suicidal ideation were generally not a clinical problem in this population.

*3.4. Other Psychological Outcomes*

Quality of life and body image were categorized as other psychological outcomes that were frequently investigated. Quality of life was assessed using the Short Form Health Survey (SF) [92,93], while body image was assessed using the Body Image Questionnaire (BIQ) [60] following recommendations from Cull on sexual function in cancer patients [94].

3.4.1. Short Form Health Survey

Eight studies assessed quality of life with some version of the SF questionnaire, with four studies utilizing the original SF-36 [44,51,53,74], three studies utilizing the SF-12 [37,39,47], and one study using two subscales of the SF-36 [62]. One study did not report outcomes relevant to the population [53]. The original SF-36 has eight subscales (vitality, physical functioning, bodily pain, general health perceptions, physical role functioning, emotional role functioning, social role functioning, and mental health) that may be summarized into a physical and a mental composite score. Two studies compared quality of life in carriers with non-carriers [44,74]. One study found lower quality of life in some domains, especially in premenopausal women (emotional role functioning, physical role functioning, and physical functioning) [44], whereas the other found no differences [74]. The other studies assessed quality of life in terms of risk management strategies. One study found that higher physical functioning was associated with higher adherence to recommended risk management strategies, but higher mental functioning was not [39]. In terms of opting for one strategy over the other, one study found no difference in quality of life between carriers opting for surgery or surveillance [37], whereas one study found that carriers with lower general health perceptions were more likely to opt for a salpingo-oophorectomy [62]. After risk-reducing surgeries, one study found lower physical quality of life six months after bilateral mastectomy but higher mental quality of life [51], whereas another study found no differences in either composite score after salpingo-oophorectomy [47].

3.4.2. Body Image Questionnaire

Four of the studies included body image as measured by the BIQ [51,55,60,78]. Two of the studies found that body image satisfaction was lower in carriers compared to non-carriers [51,78]. Longitudinally, body image satisfaction of carriers further declined as time after genetic test result disclosure passed [51,78]. One study found that body image satisfaction was unrelated to prophylactic mastectomy uptake [55]. However, two studies showed that undergoing prophylactic mastectomy, mostly combined with immediate reconstruction, may result in lower body image [51,60]. One study specified that those with lower BMI and higher cancer distress at baseline showed lower body image after finishing reconstruction, whereas higher general physical health predicted better body image over time [51]. However, it is unknown how long ago these study participants were found to carry a pathogenic variant and how that might have impacted results.

3.4.3. Summary Other Outcomes

Quality of life seemed to be largely unaffected by a positive genetic test result, although there was some evidence that especially younger women were less satisfied with their role functioning in life. It seems plausible that this was related to distress, which was also found to be slightly more prominent in premenopausal women (see Section 3.2.7). In terms of body image, the results were extremely heterogeneous and only provided limited insight. From the studies identified, it could be concluded that body image may decrease slightly after genetic test result disclosure but was generally unrelated to further decision making.

*3.5. Quality Assessment*

All the studies included in this review met at least 11 of 20 AXIS points (range: 11–20). The overall quality of the studies was adequate: most of the studies clearly stated the aims of the study, identified a clearly defined target group per inclusion criteria, and

included a good description of the basic data with justified conclusions. All but four studies discussed the limitations of the study and the results. Nonetheless, the included studies have some methodological weaknesses: most of the studies did not justify their sample size or did not run a priori power analyses. Additionally, although most studies took a sample from an appropriate frame with an appropriate sampling method, more than half (60%) of the 45 studies expressed concerns about the representativeness or indicated that a bigger sample size would have been desirable. Only 19 studies included information about non-responders, with 10 of these identifying differences between responders and non-responders. A common difference identified was that non-responders were less likely to have a partner, which is a factor to be considered in interpreting results. All AXIS results can be seen in Supplementary Material File S1.

## 4. Discussion

To the best of our knowledge, this is the first time a systematic review investigated not only the psychological outcomes of cancer-unaffected *BRCA1/2* pathogenic variant carriers, but also the instruments that were used to assess these outcomes. Due to the high heterogeneity of measures used by the different studies, it was challenging to draw comprehensive conclusions about all the psychological outcomes. The differences in the design and analyses in the presented studies may underlie this non-conclusive pattern of results.

The psychological outcomes that were most often assessed were distress, anxiety, and cancer worry. Most studies showed an increase in those outcomes, mainly cancer worry and anxiety, after genetic test result disclosure. This appeared to be slightly more prominent in premenopausal women under the age of 50 [44,47,81]. This seems logical considering family planning and breastfeeding decisions for women of childbearing age. In fact, qualitative studies with premenopausal *BRCA1/2* pathogenic variant carriers have confirmed that family planning often competes with risk-reducing surgical procedures [22,95]. This may in turn increase anxiety and distress in this younger group. Longitudinally, most studies showed a steady decline in the months after genetic test result disclosure and a complete return to baseline roughly after one year. Only a few studies reported higher frequency of distress one year after genetic test result disclosure. In terms of decision making, it seemed that women deciding for prophylactic surgeries experienced slightly higher levels of distress. This may be the reason why these women opted for risk-reducing surgeries in the first place. In terms of depressive symptomatology and quality of life, merely mild or no negative outcomes at all were identified. Regarding body image, no conclusive results could be drawn due to the small number of studies using a validated measure. Two reviews on various body image outcomes showed that decreased body image and changes in sexuality were common after prophylactic mastectomy [96,97]. However, a recent review reported that sexual health remained understudies in the context of *BRCA1/2* testing [98].

*Limitations and Recommendations for Future Research*

While this review was the first review to systematically investigate the effects of *BRCA1/2* pathogenic variant carrier status on psychological outcomes, there are a few limitations that need to be addressed. Firstly, the oldest study included in the review was published 1997, and many others were published in the early 2000s. Breast and ovarian cancer risk may have been communicated differently in those years compared to today, as they were not as well-researched and long-term data were not yet available. This may, in turn, influence the level of psychological morbidity. Secondly, the majority of the studies in the review were conducted in the United States or Europe and investigated mainly well-educated white women. Studies that specifically looked at minorities were very few. Only one study in the review examined an African-American population [56]. Thus, further and larger studies investigating such underrepresented groups are necessary. Moreover, we suspect that at least a few studies reported on the same population over several years,

which may taint the results. Two studies reported from Rambam Health Care Campus in Israel [43,44]; two further studies reported baseline and follow-up data from a sample at Rotterdam University Hospital in the Netherlands [59,60]; four studies reported from Oslo University Hospital in Norway within a close timeframe [49,50,70,71]; and finally, four studies utilized the GENESPO study cohort from France [55,58,63,64]. We were unable to exclude the possibility that more studies reported on these or other populations across different publications. Lastly, most studies reported on small study populations, with the majority of the studies including less than 100 cancer-unaffected carriers. This may impact the generalizability of our results.

As discussed above, there have been attempts to condense results from various outcome sources (e.g., integrative reviews on body image [96,97]), but the consequent and continuous use of established and validated instruments is often lacking. Future research could improve data on psychological morbidity in cancer-unaffected *BRCA1/2* pathogenic variant carriers by (1) using validated measures, (2) not conflating cancer-unaffected with cancer-affected carriers or cancer-unaffected carriers with the general population when reporting results, (3) reporting precisely how long carriers knew of their risk status when reporting results, and (4) diversifying the sample populations. Additionally, while *BRCA1/2* pathogenic variants have been known the longest and are well-studied because they are also found comparatively frequently in individuals at risk, several other pathogenic variants in less frequently identified genes exist that have similarly high risks associated with them, such as *PALB2* [99]. Future research should address these pathogenic variants equally in researching psychological morbidity in the hereditary cancer field.

**Supplementary Materials:** The following supporting information can be downloaded at: https://www.mdpi.com/article/10.3390/curroncol30040274/s1, Supplementary Material File S1: AXIS Report.

**Author Contributions:** Conceptualization, A.I. and S.S.; methodology, A.I.; software, Z.L.; validation, A.I., Z.L. and S.S.; formal analysis, A.I. and Z.L.; investigation, A.I. and Z.L.; resources, S.S.; data curation, A.I. and Z.L.; writing—original draft preparation, A.I.; writing—review and editing, Z.L., K.R. and S.S.; visualization, Z.L.; supervision, S.S.; project administration, A.I. All authors have read and agreed to the published version of the manuscript.

**Funding:** This research received no external funding.

**Conflicts of Interest:** The authors declare no conflict of interest.

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
