# Peer review of "Assessing Psychological Morbidity in Cancer-Unaffected BRCA1/2 Pathogenic Variant Carriers: A Systematic Review"

_curroncol, doi:10.3390/curroncol30040274_

Round 1

Reviewer 1 Report

1.The term ”cancer unaffected BRCA1/2 pathogenic variant carriers” was confusing. How about adding a hyphen “-” to become “cancer-unaffected BRCA1/2 pathogenic variant carriers.” Other similar terms should also adjusted accordingly.

 2. Line 129, the three authors are “AI, ZL and MR” ? or “AI, ZL and KR”?

 3. Most of the included studies had a small sample size with participants below 100, therefore, would you like to discuss results with “effect sizes” not only on basis of statistical significance? If not, it should also be included in the limitation section.

Author Response

Dear reviewer, 

thank you sincerely for reviewing our manuscript on assessing psychological burden in BRCA1/2 pathogenic variant carriers. We appreciate your helpful suggestions on how to improve our manuscript.

In the following, we are going to respond to your suggestions: 

The term ”cancer unaffected BRCA1/2 pathogenic variant carriers” was confusing. How about adding a hyphen “-” to become “cancer-unaffected BRCA1/2 pathogenic variant carriers.” Other similar terms should also adjusted accordingly

We agree with this point and have adjusted the manuscript. We have edited "cancer unaffected" to cancer-unaffected and "cancer affected to cancer-affected in the hopes of providing more clarity.

Line 129, the three authors are “AI, ZL and MR” ? or “AI, ZL and KR”?

Thank you for bringing this to our attention. We have edited the correct authors. 

Most of the included studies had a small sample size with participants below 100, therefore, would you like to discuss results with “effect sizes” not only on basis of statistical significance? If not, it should also be included in the limitation section.

Thank you for addressing this issue that we have been similarly reflecting on while writing this manuscript. Unfortunately, it is not feasible to report effect sizes as they were often not available in the original studies. Moreover, this was beyond the scope of this manuscript. We have added a paragraph in the discussion to address this limitation. 

Finally, we would like to thank you once again for your timely review and wish you all the best. 

The authors

Reviewer 2 Report

The topic of the paper is interesting and meets the aims and scope of the journal. Your systematic review is well-written. However, I would like to suggest to write a chart in which you will refer how many articles you found, how many you excluded etc.

Also, in the section of limitations you refer clearly the strengh of your study. 

Author Response

Dear reviewer, 

we thank you sincerely for reviewing our systematic review on assessing psychological burden in BRCA1/2 pathogenic variant carriers. 

We would like to address this comment made by you: "However, I would like to suggest to write a chart in which you will refer how many articles you found, how many you excluded etc."

We have indeed included such a chart on page 4 of the manuscript (figure 1). 

We have expanded on the limitations of our study in the discussion section and trust that we addressed all your points sufficiently. 

We thank you again and wish you all the best.

Kind regards

The authors